# Exploring the Interplay between the Clinical and Presumed Effect of Botulinum Injections for Cervical Dystonia: A Pilot Study

**DOI:** 10.3390/toxins15100592

**Published:** 2023-09-30

**Authors:** Harald Hefter, Sara Samadzadeh

**Affiliations:** 1Department of Neurology, University of Düsseldorf, Moorenstrasse 5, 40225 Düsseldorf, Germany; sara.samadzadeh@yahoo.com; 2Charité–Universitätsmedizin Berlin, Corporate Member of Freie Universität Berlin and Humboldt-Unverstät zu Berlin, Experimental and Clinical Research Center, 13125 Berlin, Germany; 3Department of Regional Health Research and Molecular Medicine, University of Southern Denmark, 5230 Odense, Denmark; 4Department of Neurology, Slagelse Hospital, 4200 Slagelse, Denmark

**Keywords:** cervical dystonia, natural history, long-term outcome of botulinum neurotoxin therapy, rating of improvement, presumed course of disease severity without therapy

## Abstract

Background: Repetitive intramuscular injections of botulinum neurotoxin type A (BoNT/A) are the treatment of choice in patients with cervical dystonia (CD). As soon as BoNT therapy is initiated, the natural course of CD cannot be observed anymore. Nevertheless, the present study focuses on the “presumed” course of disease severity under the assumption that no BoNT therapy had been performed. The “experienced” benefit is compared with the “presumed” worsening. Methods: Twenty-seven BoNT/A long-term-treated CD patients were recruited. They had to assess the remaining severity of CD in percent of its severity at the start of BoNT therapy (RS-%). Then, they had to draw the course of severity from the onset of symptoms to the start of BoNT/A therapy (CoDB graph), as well as the course of severity from the start of BoNT/A therapy until the day of recruitment (CoDA graph). Then, they were instructed to presume the development of CD severity from the day of the start of BoNT/A therapy until the day of recruitment under the assumption that no BoNT/A therapy had been performed, and to assess the maximal severity they could presume in percent of the severity at the start of BoNT therapy (IS-%). Then, they had to draw the “presumed” development of CD severity (CoDI graph). The “experienced” change in disease severity and the “presumed” change since the start of BoNT/A therapy were compared and correlated with a variety of demographical and treatment-related data, including the actual severity of CD at the day of recruitment, which was assessed using the TSUI score and the actual dose per session (ADOSE). Results: No CD patients expected an improvement without BoNT therapy. “Presumed” worsening ((IS-%)-100) was about 50% in the mean and did not correlate with the “experienced” benefit (100-(RS-%)). However, IS-% was significantly correlated with ATSUI and ADOSE. Conclusion: Obviously, CD patients have the opinion that their CD would have further progressed and worsened if no BoNT/A therapy had been performed. Thus, the total benefit of BoNT/A therapy for a patient with CD is a combination of the “experienced” benefit under BoNT/A therapy and the prevented worsening of CD that the patient expects to occur without BoNT/A therapy.

## 1. Introduction

Repetitive intramuscular injections of botulinum neurotoxin type A or B (BoNT/A resp. BoNT/B) have become the treatment of choice for patients with cervical dystonia (CD) [1,2]. Nowadays, more than 80% of patients with CD are treated with BoNT/A or BoNT/B [3]. Most BoNT-treated CD patients have a high adherence to therapy [4,5] and are satisfied with this treatment [3,6] in principle. Primary non-responders seem to be rare [7]. The rate of secondary treatment failures (STFs) is a matter of debate, since no clear-cut definition of secondary treatment failure exists [8,9,10,11]. Even the rate of antibody-induced secondary treatment failure (STF) is unclear, since it heavily depends on the sensitivity of the assay used to detect neutralizing antibodies (for a recent review, see [11]).

The benefit of long-term BoNT therapy rated by CD patients ranges between 0 and 100%, and can be estimated to be about 40 to 60% in the mean [7,12]. Usually, the treating physicians’ ratings of the long-term benefit of BoNT/A therapy in CD appear to be higher than patients’ ratings [7,13]. The probable reason is that CD patients suffer from symptoms and aspects of CD that the treating physicians cannot observe directly (for a more detailed discussion, see [13]). However, when symptoms of CD that can directly be observed by both patients and treating physicians are rated (e.g., an abnormal head position), a highly significant correlation is found (see [13]).

This was confirmed by comparing the benefit of BoNT therapy in CD as rated by the treating physicians and the benefit scored by long-term BoNT-treated CD patients drawing the course of disease after the start of BoNT therapy [7]. In summary, both patients and treating physicians agree on the fact that the intramuscular injection of BoNT/A or BoNT/B is an effective treatment for CD that can be performed over years with only mild side effects [7,12].

However, CD is a disease of the central nervous system (CNS), and can progress. Before the BoNT era, it was well-known that CD could worsen and that relapses were not rare [14]. The spread of CD symptoms and its progression from a focal to a multifocal state in at least 30% of patients has repeatedly been reported [15,16]. Furthermore, when CD patients are asked about the presence of special symptoms at the start of BoNT therapy in comparison to the presence of symptoms after BoNT long-term therapy, it is obvious that the spectrum of symptoms becomes broader with time [17]. Despite the improvement of some symptoms of CD under continuous BoNT treatment, other symptoms of CD may become clinically manifest [17]. Thus, BoNT injection therapy is a symptomatic therapy of CD and not a causal treatment. 

With the start of BoNT therapy, the natural course of CD can no longer be observed, neither by the patient nor by the treating physician. But both patient and physician should be aware that CD may progress. We therefore became interested in the question of what CD patients think would have happened if BoNT therapy had not been initiated. To shed some light on this question, touching upon the fears and sorrows of patients regarding the development of their disease, we again used a method involving the patients drawing their course of disease (CoD graph drawing [6,17]). 

We used this method to compare the development of disease severity before and after the start of BoNT therapy (comparison of CoDB and CoDA graphs). Compared to the drawing of the experienced course of disease, the drawing of the presumed course of disease severity without BoNT therapy (CoDI graph) was more challenging for the patients. But in the end, 27 CD patients succeeded in drawing not only a CoDB and CoDA graph, but also a CoDI graph, making it possible to perform the present pilot study comparing the “experienced” benefit (see Section 5) of BoNT therapy with the “presumed” worsening (see Section 5) of the disease without such treatment.

Our primary hypothesis was that patients with an excellent response would not suspect the severe progression of their disease: the “experienced” benefit should be negatively correlated with “presumed” worsening. Furthermore, we thought that patients would forget about the severity of CD at the start of BoNT therapy and expected the duration of treatment to have a negative influence on “presumed” worsening.

## 2. Results

### 2.1. Demographical and Treatment Related Data

For the present study, 27 patients with CD were consecutively recruited. Eighteen patients were male and nine patients were female. The duration of treatment (DURT) with botulinum neurotoxin type A (BoNT/A) was long (mean: 14.9 years, SD: 8.5 years; range: 2.5 to 33.9 years). Correspondingly, the age at recruitment (AGE) was high (mean: 68.1 years, SD: 11.9 years). However, the age at the onset of symptoms (AOS) was typical and covered a broad range (mean: 49.2 years, SD: 14.3; range: 16.4 to 73.1 years). The mean time between the first onset of symptoms to the start of BoNT/A injection therapy (DURS) was 3.89 (SD: 5.66), and ranged from 0.44 to 22.3 years. At the time of recruitment, 22 patients were treated with incobotulinum neurotoxin type A (incoBoNT/A; Xeomin^®^) and only 5 patients were treated with abobotulinum neurotoxin type A (aboBoNT/A; Dysport^®^). The mean actual total dose per session was 341 uDU (SD: 108; for the definition of unified dose units (uDU), see Section 5). In total, six out of twenty-seven patients (=22%) were classified as patients either with a primary (two patients) or secondary treatment failure (four patients). The entire group (ALL) was divided into a subgroup of these six patients with treatment failure (TF group) and the rest of the patients (n = 21; N-TF group) (see Table 1 and Figure 1 and Figure 2).

### 2.2. Long-Term Outcome, as Assessed by the Treating Physician and Patient

The long-term outcome was assessed by the treating physician using the TSUI score (ATSUI; [18]), and its mean was 3.41 (SD: 2.27). The ATSUI in the TF group was 5.33 (SD: 1.96), and was significantly (*p* < 0.024) worse than the ATSUI in the N-TF group (mean: 3.00; SD: 1.90). When patients assessed the long-term outcome by estimating the remaining severity at recruitment in percent of the severity at the start of BoNT/A therapy (RS-%), a broad spectrum of responses (5% to 160%) was obtained (comp. Figure 1A). The RS-% was significantly (*p* < 0.001) lower in the N-TF group compared to the RS-% in the TF group (details in Table 1). Also, when patients assessed their long-term outcome by drawing the remaining severity on a VAS scale in comparison to the severity at the start of BoNT/A therapy (RS-M; for details see Section 5), a significant (*p* < 0.001) difference was found (31% in the N-TF group vs. 116% in the TF group; for details, see Figure 1B and Table 1). The actual dose per session (ADOSE) was higher in the TF group (mean: 411; SD: 128) than in the N-TF group (mean: 330; SD: 86), but the difference was not significant (see Table 1).

### 2.3. Assessment of Presumed Outcome without BoNT Injection Therapy or DBS Operation

When patients presumed the possible outcome of their disease under the assumption that no BoNT injections and no DBS operation had been performed, and scored the presumed severity in % of the severity at the start of BoNT therapy, a broad spectrum of responses was observed (Figure 1C). No patients presumed an improvement. Also, when the presumed worsening had to be indicated on a VAS scale (IS-M), most of the patients presumed that their disease would worsen (Figure 1D). The IS-% and IS-M values were slightly higher in the TF group than in the N-TF group, but these differences were not significant (Figure 1D), neither for IS-% nor for IS-M (for details, see also Table 1).

### 2.4. Correlation Analysis

When a cross-correlation analysis (see Table 2) was performed for various parameters (AOS, AOT, AGE, DURS, DURT, RS-%, RS-M, IS-%, IS-M, ADOSE, ATSUI, 100-(RS-%), (IS-%)-100), some trivial and some less obvious correlations were detected. The highly significant correlations observed among the age parameters (AOS, AOT, AGE) can be categorized as trivial results. However, the highly significant correlation (r = 0.50, *p* < 0.008) between DURS and ATSUI is less trivial, as is the correlation between the treating physician’s rating and patient’s assessment (ATSUI vs. RS-%: r = 0.56, *p* < 0.003). Correspondingly, DURS and the patient’s rating also yield significant correlations (RS-%: r = 0.72, *p* < 0.001; RS-M: r = 0.73, *p* < 0.001). Furthermore, ATSUI is significantly correlated (r = 0.56, *p* < 0.003) with ADOSE. 

Patients with a higher ATSUI presume a higher level of possible worsening without therapy (IS-%: r = 0.50, *p* < 0.008; IS-M: r = 0.54, *p* < 0.003). As a result of the correlation between ATSUI and ADOSE, IS-% is also significantly correlated (r = 0.43, *p* < 0.009) with ADOSE.

However, there is no significant correlation between RS% with respect to RS-M and IS-% or IS-M (RS-% vs. IS-%: r = 0.27, *p* = 0.167, n.s. and RS-M vs. IS-M: r = 0.27, *p* = 0.167, n.s.). This can also clearly be seen in Figure 2A, where the “experienced” benefit 100-(RS-M) (Figure 2, x-axis) is plotted against the “presumed” worsening (IS-M)-100 (Figure 2A, y-axis). There is a negative slope in the regression line, which is similar between the TF group (black dots) and the N-TF group (grey dots). The overall regression line is not significant (r = 0.27, *p* = 0.167).

## 3. Discussion

### 3.1. Reasons for the Excellent Outcome in the Present Cohort

The present cohort is exceptional in several ways. To our knowledge, its mean duration is the longest reported so far. Furthermore, the mean outcome (measured using the TSUI score [18]) is 3.4 +/− 2.21, which is exceptionally low compared to other studies using the TSUI score as an outcome measure ([19,20], for further comparisons and an overview, see also Figure 8.2 and Table 8.2 in [12]). The long-term mean TSUI score in a large cross-sectional study was 4.75 +/− 3.2 in 173 CD patients when using a negative mouse hemidiaphragma (MHDA) antibody (AB) test, and 6.29 +/− 3.8 in 31 MHDA-positive CD patients [21]. When the 21 patients without TF (N-TF group) in the present study assessed the outcome, a mean improvement of around 70% after a mean duration of about 15 years (compare with Table 8.2 in [12]) was found. In this subgroup, physicians rated an exceptionally low mean TSUI score of 3.0 (see Table 1).

During the treatment of CD with repetitive injections every 3 months, such low TSUI-scores are usually observed after a treatment duration of about 1000 days. This was detected when we looked for the lowest TSUI score (best TSUI score; BTSUI) during the treatment of a patient. A mean BTSUI of 3.75 was observed after a treatment period of 1354 days (corresponding to 15 injections), with a mean dose of 843 U aboBoNT/A (=281 uDU). The mean BTSUI was 3.83 in a group of CD patients exclusively treated with 321 U onabotulinum neurotoxin type A(onaBoNT/A; Botox^®^) over 1085 days (corresponding to 12 injections), and was 1.71 in a group of CD patients exclusively treated with 267 U incoBoNT/A over 920 days (corresponding to 10 injections) [22]. Having reached its best value, the TSUI score slowly increases again when the dose of BoNT/A is not increased. With the increase in dose, the outcome (TSUI score) can be kept on a fairly low level. In the patients without TF, a mean dose of 330 +/− 86 uDU was applied, which is well above the doses reported in other studies using the TSUI score as an outcome measure (for a comparison, see also Table 8.2 in [12]). Even in the TF group in the present study, in whom the dose was increased up to 411 +/− 128 uDU, the mean outcome (mean TSUI score: 5.33, SD: 196) was better than in the MHDA-positive patients in [21], who were treated with 266 +/− 39 uDU.

In summary, the good outcome observed in the present study can be explained, to a large extent, by the exceptionally high doses that were used to compensate the slow progression of the severity of CD over time during treatment.

### 3.2. Realistic Estimation of the Progress of CD without Specific Therapy

When BoNT therapy is initiated, the natural course of CD can no longer be observed. We hypothesized (see Introduction) that patients with a good long-term outcome would presume that a lower degree of CD impairment would develop without intervention (as BoNT injection therapy or DBS operation) than patients with a worse long-term outcome. We expected a negative correlation between “presumed” worsening and “experienced” improvement, and between age or treatment duration and “presumed” worsening (because patients would probably forget about the severity of their CD 15 years ago). 

Interestingly, no patients expected an improvement without BoNT therapy (Figure 1). Our experience is that patients usually come to therapy when their TSUI score exceeds 8 (compare with [12,19]). Less than 25% of de novo CD patients present with a TSUI score larger than 10. CD patients with an initial TSUI score larger than 14 are rare (<5% [19]). Thus, a TSUI score of 14 to 16 demarcates the usual upper limit of severity of CD in de-novo CD patients. 

More than 80% of the CD patients without TF in the present cohort (17/21) presumed a worsening of less than 170%. By using a TSUI score of 10 to correspond to a severity of 100% at the start of BoNT therapy, this would imply that most of the patients imagined that their TSUI score would not exceed a value of 16. We therefore think that the presumption of the patients was realistic.

A correlation analysis between IS-% or IS-M and the duration of treatment or AGE did not yield a significant relation. Also, the correlation between the patient′s assessment of improvement or “experienced” benefit 100-(RS-M) and “presumed” worsening (IS-M)-100 did not show a significant negative relation (see Table 2 and Figure 2A). Thus, in the present small cohort of long-term-treated CD patients, the patient′s rating of their long-term outcome and duration of treatment did not significantly influence the “presumed” worsening or “presumed” disease progression of CD. Patients who had developed a TF expected a significantly worse outcome compared to patients without TF (Figure 1), but their assessment of “presumed” worsening was not significantly different from that of the patients without TF (Figure 2A). However, there was a significant positive correlation between ATSUI and IS-% (or IS-M). 

There was a significant (*p* < 0.001) negative correlation between the time to therapy (DURS) and long-term outcome when rated by the treating physician (ATSUI) or rated by the patient (RS-% with respect to RS-M). This has also been reported for the long-term outcome of patients with primary dystonia after DBS operation [23,24].

### 3.3. Consequences of an “Presumed” Progression of CD without Specific Therapy

To semi-quantify the effect of the various scales employed for BoNT injections, such as the TSUI score [18], which is used in the present study, the Toronto Western Spasmodic Torticollis scale (TWSTRS) [25], the Burke–Fahn–Marsden scale [26], the craniocervical dystonia questionnaire (CDQ24) [27] and others (see [28]) have been introduced. Usually, these scales are used to quantify the peak effect of a single injection (e.g., in [29]).

However, in daily practice and to determine the long-term outcome and whether antibodies may have been induced, it is important to know whether the duration of the effect of a single injection is preserved or not [30]. When patients are injected every 3 months, it is important to score them immediately before the next injection because a decline in the score corresponds to a decline in the duration of the clinical effect. If the clinical effect declines, the patient will experience a situation similar to that experienced before BoNT therapy, but with a less positive perspective. Thus, patients do not only rate the actual improvement realistically, but also the possible presumed worsening, as can be seen from the significant correlation between ATSUI and IS-% (and IS-M).

We therefore think that the most important implication of the “presumed” progression of CD is that the benefit of BoNT therapy for a patient is a combination of the “experienced” improvement in CD and the prevention of the expected worsening of CD. This is the most likely reason for the high adherence to therapy in BoNT-treated patients in general [5]. At the end of each injection cycle, patients realize that a stable plateau of improvement can only be achieved when the BoNT injections are performed repetitively and regularly.

A second implication is that patients with CD have a realistic experience of the underlying progression of CD. It was repeatedly reported by patients in our cohort during the CoDI graph drawing that the spectrum of symptoms had become broader, although the BoNT injections were considered to be very helpful in tolerating individual symptoms (see [17]). 

A third implication is that a worsening reported by a patient does not necessarily imply that antibodies have been induced and that a secondary treatment failure is developing. Instead, the dose and injection scheme should be adapted to the worsening of symptoms, the development of new symptoms, or the spread of symptoms to nearby parts of the body [15,16,31,32]. We think that this is performed in most BoNT centers and is the reason why an increase in the dose with the duration of treatment is reported in most long-term studies (e.g., [33,34]).

## 4. Conclusions

When patients with CD under long-term BoNT treatment are asked to presume the possible development of CD if no intervention is performed, such as BoNT injection therapy or DBS operation, they document a possible worsening that lies in the upper or maximal range of severity observed in de novo CD patients who come to be injected with BoNT. This “presumed” worsening is not influenced significantly by the “experienced” benefit under BoNT therapy. We therefore think that CD patients under BoNT therapy not only experience an improvement in a variety of symptoms, but also experience the onset of new symptoms, indicating the spreading out of symptoms and the progression of CD. 

### Strengths and Limitations of the Study

The strength of this study is that, to our knowledge, this is the first study to analyze the “presumed” worsening of CD in BoNT long-term-treated CD patients. Although we are convinced that the “presumed” worsening of our patients was realistic, it has to kept in mind that this study delt with subjective data. Therefore, these data should be compared to data from CD patients who have not been treated with BoNT injections. Unfortunately, such patients were not available in our out-patient department. A multi-center study should be initiated to collect more information on the natural history of CD for comparison with BoNT- or DBS-treated CD patients.

## 5. Materials and Methods

The inclusion criteria for the present study were as follows: (i) patients whose diagnosis of CD was confirmed in the out-patient department of Neurology at the University of Düsseldorf, and (ii) patients who had undergone continuous treatment for CD for at least two years. The exclusion criteria were as follows: (i) patients whose BoNT treatment had been ceased for more than 6 months (corresponding to two treatment cycles); (ii) patients with memory problems such as Alzheimer’s disease; (iii) patients with treated psychiatric disorders, e.g., moderate to major depression; and (iv) CD patients with other BoNT-treated diseases, such as migraine, bruxism or neuroinflammatory diseases.

Patients were informed about the purpose of the present study while they were waiting for their next routine BoNT/A injection in the out-patient department of Neurology at the University of Düsseldorf. After the patients had given informed consent, they were recruited for the present study and received their BoNT injection.

Patients were classified as patients with primary treatment failure (PTF) when their improvement during the entire course of treatment (assessed using the TSUI scale [19]) was lower than 3 points. Patients were classified as secondary non-responders (STF) when they experienced an improvement of at least 3 TSUI points, followed by a systematic worsening of at least 2 TSUI points over at least two treatment cycles during their course of treatment (for details of the definition of secondary non-responders, see [22]). Two patients were primary non-responders (open symbols in Figure 1 and Figure 2), and four patients were secondary non-responders (full black symbols in Figure 1 and Figure 2). These 6 patients were collected in a separate treatment failure (TF) subgroup, in contrast to the 21 other patients, whose treatment had not clinically appeared to fail (N-TF subgroup).

After the injection, the patients had to remember when they had noticed symptoms of CD for the first time and had to assess the actual remaining severity of CD (RS-%) in percent of the severity of symptoms at the start of BoNT therapy (=100%). 

Then, they had to draw (i) the course of CD severity from the onset of symptoms to the start of BoNT therapy (CoDB graph), (ii) the course of CD severity from the start of BoNT therapy to recruitment (CoDA graph), and (iii) the presumed course of disease severity from the start of BoNT therapy until recruitment (CoDI graph) under the assumption that no intervention (neither BoNT therapy nor DBS operation) had been performed (see Figure 3).

For the CoDB graph drawing, the patients were comfortably seated in front of a table. A sheet of paper with a square size of 10 × 10 cm (see Figure 3; left part) was placed in front of the patients. It was explained to the patients that the lower left corner represented the onset of symptoms and that the upper right corner represented the severity of CD at the start of BoNT therapy. The patients had to draw a continuous line from the lower left to the upper right corner (CoDB graph), representing the course of severity of the disease before BoNT therapy. Three attempts were allowed, but no drawing assistance was permitted. To avoid any bias, the patients were not shown an example.

For the CoDA graph drawing (Figure 3, right part), a second sheet of paper with a square of 10 × 10 cm was placed in front of the patients. This time, the upper left corner represented the severity of CD at the start of BoNT therapy. The patients had to assess the actual remaining severity of CD (RS-%) in percent of the severity of CD at the start of BoNT therapy, and then had to mark RS-% on the right edge of the square (RS-M; thus, the right edge of the square was used as a visual analogue scale (VAS)). Then, the patients had to draw a continuous line from the upper left corner of the square (=100%) to the RS-M mark (CoDA graph). Again, three attempts were allowed, and no example was shown. 

In a third step, a third sheet of paper was presented to the patients with a right-angle of 10 × 22 cm size. On the left edge, a mark was added at 10 cm, representing the severity of CD at the start of BoNT therapy (=100%). Then, the patient had to presume how the severity of CD would have possibly developed, from starting BoNT therapy until recruitment, if no BoNT treatment or DBS operation had been performed; they then had to assess the maximal “presumed” disease severity in percent of the severity at the start of BoNT therapy (IS-%). Then, the patient had to mark IS-% on the right edge of the rectangle (IS-M), and then draw a continuous graph from the 100% mark on the left edge to the IS-M mark, corresponding to the presumed severity of CD from the start of BoNT therapy until the day of recruitment (CoDI graph). Three attempts were allowed, but no example was presented.

In the present study, only four parameters (RS-%, RS-M, IS-%, IS-M) are analyzed. The shape of the different CoD graphs will be analyzed in a subsequent paper when the number of patients is large enough for the cohort to be further subdivided. The parameters 100-(RS-%) and 100-(RS-M) assess the “experienced” benefit, the parameters (IS-%)-100 and (IS-M)-100 assess the “presumed” worsening, and the sums (IS-%)-(RS-%) and (IS-M)-(RS-M) assess the total benefit (TBEN), which is presented in detail in Figure 2B. 

For further analysis, the following parameters were extracted from the charts of the patients: AGE, date of the start of BoNT therapy, actual severity of CD, as assessed by the treating physician using the TSUI scale (ATSUI), actual used BoNT/A preparation and actual total dose of BoNT/A per session (ADOSE). For the sake of comparison, doses were transformed into unified dose units (uDU) by dividing the aboBoNT/A doses by three and leaving the ona- and incoBoNT/A doses unchanged (following a consensus paper [35]). Calculated were the age at the onset of symptoms (AOS), age at the start of BoNT therapy (AOT), time between the onset of symptoms and the start of BoNT therapy (DURS), and the duration of therapy (DURT = time span between the start of BoNT therapy and recruitment). 

### Statistics

Mean values were non-parametrically compared using the Mann–Whitney U test. The parameters of the TF and N-TF group were compared using the Wilcoxon Rank Sum test. Finally, the non-parametric rank correlation coefficients were calculated in a cross-correlation matrix between AOS, AOT, AGE, DURS, DURT, RS-%, RS-M, IS-%, IS-M, ADOSE, ATSUI and “experienced” benefit 100-(RS-%), as well as 100-(RS-M) and “presumed” worsening (IS-%)-100, and (IS-M)-100. All statistical procedures were part of the R software statistics package (version 4.3.1). The dplyr package was utilized for data manipulation, and the ggplot2 package was used for data visualization (histograms, box plots).

## Figures and Tables

**Figure 1 toxins-15-00592-f001:**
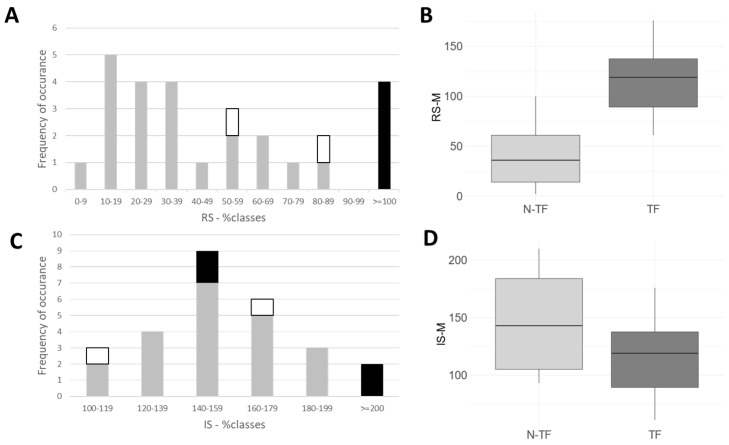
(**A**) Distribution of patient’s assessment of long-term outcome (RS-%) in the N-TF group (grey bars) and the TF group (open bars = patients with PTF, black bars = patients with STF). (**B**) Box plots of patients’ assessments of long-term outcome (RS-M) in the N-TF group and in the TF group. (**C**) Distribution of patients’ assessments of “presumed” maximal severity of CD without BoNT therapy in the N-TF group (grey bars) and in the TF group (open bars = two patients with PTF, black bars = four patients with STF). (**D**) Box plots of patients’ assessments of “presumed” outcome (IS-M) without BoNT therapy in the N-TF group and in the TF group.

**Figure 2 toxins-15-00592-f002:**
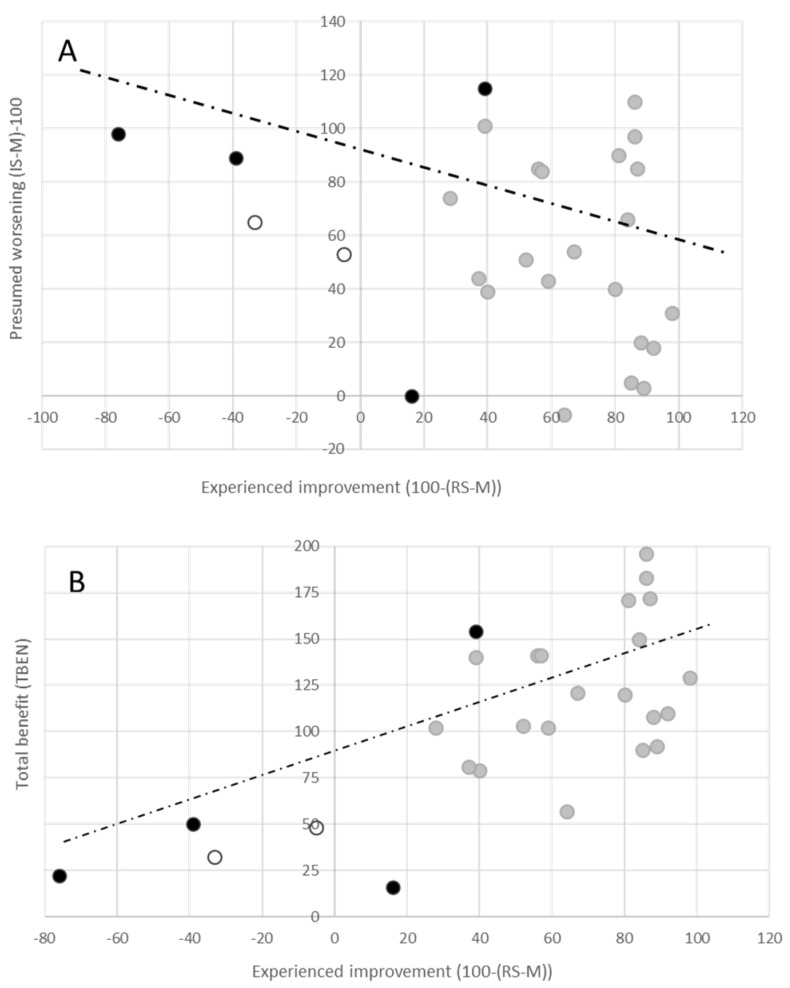
(**A**) Missing correlation between “experienced” benefit (100-(RS-M); x-axis) and the “presumed” worsening ((IS-M)-100); y-axis). Grey dots represent patients in the N-TF group, and black and open dots represent patients with STF or PTF, respectively. The line indicates the regression line between the “experienced” benefit and “presumed” worsening of the entire cohort. (**B**) Significant correlation between “experienced” benefit (100-(RS-M); x-axis) and the total benefit (TBEN: (IS-M)-RS-M; y-axis). Grey dots represent patients in the N-TF group, and black and open dots represent patients with STF or PTF, respectively. Full line indicates the regression line between the “experienced” benefit and total benefit TBEN of the entire cohort. In (**B**), the total benefit (TBEN = (IS-M) − (RS-M)), which we have defined as the sum of the “experienced” benefit (100-(RS-M)) and the “presumed” worsening ((IS-M)-100), is plotted against the “experienced” benefit. All patients have a positive benefit (even the patients with a PTF (open symbols) and a STF (full symbols). This probably explains why the patients with TF are still treated with BoNT therapy despite worsening since the start of BoNT therapy. The correlation between “experienced” and TBEN is highly significant (r = 0.7083; *p* < 0.001).

**Figure 3 toxins-15-00592-f003:**
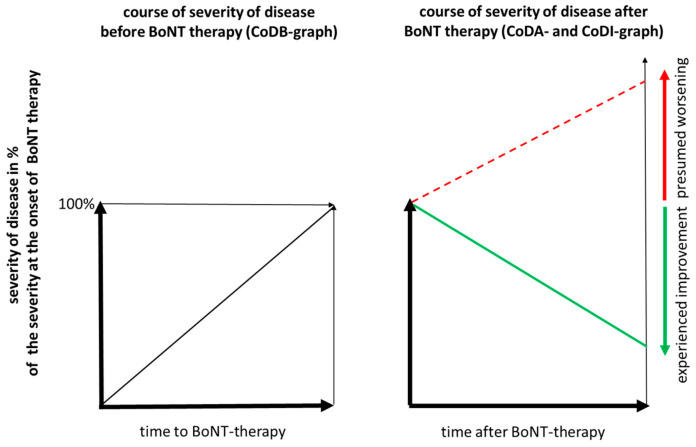
Schematical graph showing how the patients had to draw the course of disease severity before BoNT therapy (**left** part) and after BoNT therapy (**right** part; full line), and how they had to indicate the presumed worsening (**right** part; hatched line) of their disease during BoNT treatment under the assumption that no BoNT therapy and no DBS operation had been performed.

**Table 1 toxins-15-00592-t001:** Demographical and treatment-related data (when the entire group (ALL) are split up into patients with TF (TF group) and patients without TF (N-TF group).

Parameter	TF Group	N-TF Group	ALL	Wilcoxon *p*-Value
N=	6	21	27	
SEX (f/m)	2/4	7/14	9/18	
AOS (years)	40.8/12.9	51.7/13.5	49.2/14.3	0.175
AOT (years)	48.7/12.4	54.1/13.4	53.1/12.7	0.670
AGE (years)	62.5/4.5	69.8/12.5	68.1/11.9	0.097
DURS (years)	9.00/10.00	2.43/2.16	3.89/5.66	0.252
DURT (years)	12.6/5.2	15.5/9.0	14.9/8.5	0.798
ATSUI	5.33/1.96	3.00/1.90	3.41/2.21	0.024
ADOSE (uDU)	411/128	330/86	341/108	0.202
RS-%	105/37.1	29.0/19.4	46.0/37.7	0.001
RS-M	116/40.6	31.0/20.6	50.0/41.7	0.0009
IS-%	164/37.2	147/21.9	151/26.3	0.239
IS-M	170/40.9	154/33.9	158/35.1	0.307

TF = treatment failure (either primary TF or secondary TF (for details, see Section 5)), TF group = subgroup of patients with TF, N-TF group = subgroup of patients without TF, f = female, m = male, AOS = age at onset of symptoms, AOT = age at start of therapy, AGE = age at day of recruitment, DURS = time from onset of symptoms to therapy (DURS = AOT − AOS), DURT = duration of therapy, ATSUI = actual TSUI score at day of recruitment, ADOSE = actual dose at day of recruitment, RS-% = remaining severity of CD assessed by the patient in % of the severity at start of BoNT therapy, RS-M = mark of RS-% in the CoDA graph, IS-% = presumed maximal severity of CD under the assumption of no BoNT therapy, IS-M = mark of IS-% in the CoDI graph, uDU = unified dose units (see Section 5).

**Table 2 toxins-15-00592-t002:** Cross-correlation between demographical data, treatment-related data and outcome measures.

	AOS	AOT	AGE	DURS	DURT	RS-%	RS-M	IS-%	IS-M	ADOSE	ATSUI	100-(RS-%)	(IS-%)-100
AOS		r = 0.91 *	r = 0.67 *	r = −0.44 *	r = −0.38 *	r = −0.19	r = −0.19	r = −0.08	r = 0.14	r = −0.35	r = −0.27	r = 0.19	r = −0.08
AOT	*p* = 0.000		r = 0.76 *	r = −0.04	r = −0.38	r = 0.11	r = 0.13	r = 0.04	r = 0.27	r = −0.36	r = −0.07	r = −0.11	r = 0.04
AGE	*p* = 0.000	*p* = 0.000		r = 0.04	r = 0.31	r = 0.02	r = 0.02	r = 0.01	r = 0.14	r = −0.34	r = 0.07	r = −0.02	r = 0.01
DURS	*p* = 0.022	*p* = 0.858	*p* = 0.860		r = 0.10	r = 0.72 *	r = 0.73 *	r = 0.30	r = 0.25	r = 0.04	r = 0.50 *	r = −0.72 *	r = 0.30
DURT	*p* = 0.050	*p* = 0.053	*p* = 0.116	*p* = 0.604		r = −0.13	r = −0.16	r = −0.05	r = −0.18	r = 0.05	r = 0.21	r = 0.13	r = −0.05
RS-%	*p* = 0.331	*p* = 0.584	*p* = 0.905	*p* = 0.000	*p* = 0.524		r = 0.99 *	r = 0.27	r = 0.23	r = 0.21	r = 0.56 *	r = −1.00 *	r = 0.27
RS-M	*p* = 0.354	*p* = 0.531	*p* = 0.936	*p* = 0.000	*p* = 0.418	*p* = 0.000		r = 0.31	r = 0.27	r = 0.18	r = 0.54 *	r = −0.99 *	r = 0.31
IS-%	*p* = 0.678	*p* = 0.827	*p* = 0.952	*p* = 0.124	*p* = 0.813	*p* = 0.167	*p* = 0.119		r = 0.92 *	r = 0.49 *	r = 0.50 *	r = −0.27	r = 1.00 *
IS-M	*p* = 0.496	*p* = 0.181	*p* = 0.471	*p* = 0.209	*p* = 0.361	0.250	*p* = 0.169	*p* = 0.000		r = 0.30	r = 0.43 *	r = −0.23	r = 0.92 *
ADOSE	*p* = 0.078	*p* = 0.062	*p* = 0.081	*p* = 0.832	*p* = 0.820	*p* = 0.293	*p* = 0.380	*p* = 0.009	*p* = 0.123		r = 0.55 *	r = −0.21	r = 0.49 *
ATSUI	*p* = 0.178	*p* = 0.721	*p* = 0.719	*p* = 0.008	*p* = 0.292	*p* = 0.003	*p* = 0.003	*p* = 0.008	*p* = 0.025	*p* = 0.003		r = −0.56 *	r = 0.50 *
100-(RS-%)	*p* = 0.331	*p* = 0.584	*p* = 0.905	*p* = 0.000	*p* = 0.524	*p* = 0.000	*p* = 0.000	*p* = 0.167	*p* = 0.250	*p* = 0.293	*p* = 0.003		r = −0.27
(IS-%)-100	*p* = 0.678	*p* = 0.827	*p* = 0.952	*p* = 0.124	*p* = 0.813	*p* = 0.167	*p* = 0.119	*p* = 0.000	*p* = 0.000	*p* = 0.009	*p* = 0.008	*p* = 0.167	

In the upper part (above the diagonal), the non-parametric rank correlation coefficients are presented, and in the lower part, the corresponding *p*-values are presented. Grey boxes indicate significant correlations. AOS = age at onset of symptoms, AOT = age at start of therapy, AGE = age at day of recruitment, DURS = time from onset of symptoms to therapy (DURS = AOT-AOS), DURT = duration of therapy, RS-% = remaining severity of CD assessed by the patient in % of the severity at start of BoNT therapy, RS-M = mark of RS-% in the CoDA-graph, IS-% = presumed maximal severity of CD under the assumption of no BoNT therapy, IS-M = mark of IS-% in the CoDI graph, ATSUI = actual TSUI score at day of recruitment, ADOSE = actual dose at day of recruitment, uDU = unified dose units (see Section 5), 100-(RS-%) = experienced benefit, (IS-%)-100 = presumed worsening. *The green cells correspond to significant values.

## Data Availability

Data are available upon request related to the restrictions of privacy or ethics. The data presented in this study are available upon request from the corresponding author.

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
