# Peer review of "Exploring the Interplay between the Clinical and Presumed Effect of Botulinum Injections for Cervical Dystonia: A Pilot Study"

_toxins, 2023, doi:10.3390/toxins15100592_

Round 1

Reviewer 1 Report

The study "On the relation between "experienced" benefit and "imaging" worsening in botulinum neurotoxin long-term therapy of cervical dystonia: A pilot study", is a theoretical, hypothetical and psychological study, trying to enter to the patient's perception on how his/her disease will look like without repetitive botulinum injections. Despite the novelty of the study to let the patients presume the course of the disease, there are some concerns that should be addressed:

Title:

The title is long and not attractive to the potential reader. I would suggest something like: "Exploring the interplay between clinical and presumed effect of botulinum injections for cervical dystonia: A pilot study". You might suggest another alternative but the current one is not inviting.

In addition, I think that the word "imagined" is not proper. I think that "presumed" is more relevant.

Methods:

It is unclear how the authors defined "secondary treatment failure". How patients with treatment failure kept coming, even though that at least 4 of whom seem to have no effect at all from the treatment, according to figure 1A. It is unclear what kind of "patients with pain syndrome" were excluded as many CD patients are coming to the clinic with nuchal pain as the most prominent complaint.

Results:

In Table 1, STF should be clariefied in the abbreviation.

The letters in Figure 1 are much too small.

In section 2.3, the authors refer the reader to Figure 2C but it is Figure 1C. Also the other references to this Figure are wrongly given.

The p-value of the comparison between the left and right graphs should better be given in the figure.   

In section 3.3, the reference 27 is wrongly given and does not make sense, after following a thorough reading of the paper, and I recommend to replace it by the article of Yahalom et al. which gives a correct view of the statement of the authors regarding the dose increase over time (Botulinum Injections for Idiopathic Cervical Dystonia: a Longitudinal Study. Yahalom G, Fay-Karmon T, Livneh V, Israeli-Korn S, Ephraty L, Hassin-Baer S. Neurotox Res. 2021 Aug;39(4):1352-1359. )

And last and least, a few small comments:

Page 6, line 195: the word "be" should be inserted between can and explainned.

Page 6, line 199: A coma should be placed after "initiated".

Page 7, line 268: the word "diagnose" is correct in Dusseldorfian dialect but should be replaced by "diagnosis".

Page 7, line 268-269: The sentence should be as follows: "diagnosis of CD confirmed in the outpatient department of Neurology…". So is it in line 276.

The paper is well written although there are some typo errors. You can see my comments on these errors.

Author Response

The study "On the relation between "experienced" benefit and "imaging" worsening in botulinum neurotoxin long-term therapy of cervical dystonia: A pilot study", is a theoretical, hypothetical and psychological study, trying to enter to the patient's perception on how his/her disease will look like without repetitive botulinum injections. Despite the novelty of the study to let the patients presume the course of the disease, there are some concerns that should be addressed:

Title:

The title is long and not attractive to the potential reader. I would suggest something like: "Exploring the interplay between clinical and presumed effect of botulinum injections for cervical dystonia: A pilot study". You might suggest another alternative but the current one is not inviting.

In addition, I think that the word "imagined" is not proper. I think that "presumed" is more relevant.

Methods:

It is unclear how the authors defined "secondary treatment failure".

How patients with treatment failure kept coming, even though that at least 4 of whom seem to have no effect at all from the treatment, according to figure 1A.

It is unclear what kind of "patients with pain syndrome" were excluded as many CD patients are coming to the clinic with nuchal pain as the most prominent complaint.

Results:

In Table 1, STF should be clariefied in the abbreviation.

The letters in Figure 1 are much too small.

In section 2.3, the authors refer the reader to Figure 2C but it is Figure 1C. Also the other references to this Figure are wrongly given.

The p-value of the comparison between the left and right graphs should better be given in the figure.  

In section 3.3, the reference 27 is wrongly given and does not make sense, after following a thorough reading of the paper, and I recommend to replace it by the article of Yahalom et al. which gives a correct view of the statement of the authors regarding the dose increase over time (Botulinum Injections for Idiopathic Cervical Dystonia: a Longitudinal Study. Yahalom G, Fay-Karmon T, Livneh V, Israeli-Korn S, Ephraty L, Hassin-Baer S. Neurotox Res. 2021 Aug;39(4):1352-1359. )

And last and least, a few small comments:

Page 6, line 195: the word "be" should be inserted between can and explainned.

Page 6, line 199: A coma should be placed after "initiated".

Page 7, line 268: the word "diagnose" is correct in Dusseldorfian dialect but should be replaced by "diagnosis".

Page 7, line 268-269: The sentence should be as follows: "diagnosis of CD confirmed in the outpatient department of Neurology…". So is it in line 276.

The authors follow reviewer 1 and have modified the title according to his suggestion.

We follow reviewer 1 and have replaced “imagined” by “presumed”.

We now explicitly mention our definition of “secondary treatment failure” in CD we have presented more than 10 years ago.

The authors are thankful for this aspect. We think that the present paper provides a clear answer: These patients come because the total benefit (IS-M)-(RS-M) is still positive in these patients. In other words the presumed worsening is still larger than the experienced worsening. This is explained in more detail.

We are thankful to this comment. We have patients who are BoNT treated because of CD and migraine or bruxism. These patients were excluded since it may be very difficult for these patients to assess the course of CD severity. This is specified now.

This is corrected.

We agree and have increase the size of letters.

Reviewer 1 is right. We now have modified Fig. 2 and present two parts A and B which now are mentioned in the text correctly.

This is a misunderstanding: Fig. 1 does not consist of 2 parts. We distinguish between the STF-patients and the N-STF-patients, but the regression line is calculated for the entire group. The regression lines of the two subgroups are only mentioned in the text now.

We have added a part B in the Fig. 2 presenting the total benefit TBEN in more detail.

We follow reviewer 1.

This is corrected.

This is corrected.

This is corrected.

We follow reviewer 1.

Reviewer 2 Report

Dear Editor,

The manuscript entitled “on the relation between experienced benefit and imagined worsening in botulinum neurotoxin long-term therapy of cervical dystonia: a pilot study” is well written and easy to read. Some typos can be removed during the proofreading. 

My primary concern is on the experimental design. Although this methodology has already been used in another paper published in Toxins, I find the study challenging to replicate. I also have several difficulties understanding the advantages of this approach in describing a disease. If interviewed again, the same enrolled cohort could draw a different graph only because of their emotional status. Although the manuscript refers to a pilot study, the number of observed patients is low to draw some conclusions. In addition, I have several concerns about Toxins readers' interest in this manuscript. For these reasons, I think you should reject it.

Some typos can be removed during the proofreading. 

Author Response

Dear Editor,

The manuscript entitled “on the relation between experienced benefit and imagined worsening in botulinum neurotoxin long-term therapy of cervical dystonia: a pilot study” is well written and easy to read. Some typos can be removed during the proofreading.

My primary concern is on the experimental design. Although this methodology has already been used in another paper published in Toxins, I find the study challenging to replicate.

I also have several difficulties understanding the advantages of this approach in describing a disease.

If interviewed again, the same enrolled cohort could draw a different graph only because of their emotional status.

Although the manuscript refers to a pilot study, the number of observed patients is low to draw some conclusions.

In addition, I have several concerns about Toxins readers' interest in this manuscript.

For these reasons, I think you should reject it.

The methodology of the present study is very simple. We only use the assessment of the presumed or experienced severity of CD either assessed in precent of the severity at onset of BoNT-therapy (RS-%; IS-%) or marked on a visual analogue scale (RS-M; IS-M). The analysis of the course of disease graphs is postponed until the number of graphs is large enough to analyse also the form of the graphs. This is ongoing work.

We have presented evidence in another paper that the course of disease (whether it develops rapidly or slowly e.g.) has implications for BoNT-therapy and long-term outcome.

Reviewer 2 is right. The method of CoD-graph drawing has to be evaluated. Our experience is that the principle form of a CoD-graph is replicated quite well and has little to do with the emotional status. For example, patients remember quite well which handicaps had developed after manifestation of CD and when they could go to work again.

Reviewer 2 is right. The number of patients is small. But it is large enough to test the hypothesis that rating of the presumed worsening does not decrease with duration of therapy or improvement during BoNT therapy.

As presented in the discussion this approach to ask for presumed worsening touches the problem of sorrows and fears of the patients which should be of high interest to all treating physicians.

Reviewer 3 Report

The manuscript makes a dual impression. On the one hand, the material is presented in a scientific manner, with a sufficient number of observations, analyses, and discussions, and looks quite convincing. On the other hand, the use of subjective criteria in the patients' assessment of their own condition leads to a lot of unreliable data. Considering all the pros and cons, I think this article has the right to be published. Especially considering that the authors describe all the limitations of the study in detail. An additional discussion of the selected parameters for the assessment of the patient status could strengthen the manuscript.

Author Response

The manuscript makes a dual impression. On the one hand, the material is presented in a scientific manner, with a sufficient number of observations, analyses, and discussions, and looks quite convincing.

On the other hand, the use of subjective criteria in the patients' assessment of their own condition leads to a lot of unreliable data.

Considering all the pros and cons, I think this article has the right to be published. Especially considering that the authors describe all the limitations of the study in detail.

An additional discussion of the selected parameters for the assessment of the patient status could strengthen the manuscript.

In principle, reviewer 3 touches an interesting scientific discussion: do patients´ assessment necessarily lead to a lot of unreliable data. Most of our patients have scientific background as teachers, engineers or technicians and are able to assess a problem at least as good as a physician. Furthermore, nobody knows the trouble of the patient better than the patient himself.

We would like to see a well-designed study to this problem.

The authors are thankful for that comment.

As mentioned above we now mention why we focus on the four parameters RS-%, RS-M, IS-%, IS-M in the present study and discuss in more detail why these parameters were selected.

Round 2

Reviewer 2 Report

The authors revised the manuscript; however, their reply to my concerns didn't convince me and suggested the rejection. 

Some typos can be removed during the editing steps.